# Circulating Tumor Cells: Origin, Role, Current Applications, and Future Perspectives for Personalized Medicine

**DOI:** 10.3390/biomedicines12092137

**Published:** 2024-09-20

**Authors:** Maria Cristina Rapanotti, Tonia Cenci, Maria Giovanna Scioli, Elisa Cugini, Silvia Anzillotti, Luca Savino, Deborah Coletta, Cosimo Di Raimondo, Elena Campione, Mario Roselli, Sergio Bernardini, Luca Bianchi, Anastasia De Luca, Amedeo Ferlosio, Augusto Orlandi

**Affiliations:** 1Anatomic Pathology, Department of Integrated Care Processes, University of Rome Tor Vergata, Viale Oxford 81, 00133 Rome, Italy; tonia.cenci@ptvonline.it (T.C.); scioli@med.uniroma2.it (M.G.S.); anzillottisilvia@gmail.com (S.A.); luca.savino@ptvonline.it (L.S.); ferlosio@med.uniroma2.it (A.F.); orlandi@uniroma2.it (A.O.); 2Department of Experimental Medicine, University of Rome Tor Vergata, Via Montpellier 1, 00133 Rome, Italy; elisa.cgn@gmail.com (E.C.);; 3Oncology Unit, Department of Systems Medicine, University of Rome Tor Vergata, Viale Oxford 81, 00133 Rome, Italy; deborah.coletta@ptvonline.it (D.C.); mario.roselli@uniroma2.it (M.R.); 4Dermatology Unit, Department of Systems Medicine, University of Rome Tor Vergata, Via Montpellier 1, 00133 Rome, Italy; cosimodiraimondo@gmail.com (C.D.R.); elena.campione@uniroma2.it (E.C.); luca.bianchi@uniroma2.it (L.B.); 5Department of Biology, University of Rome Tor Vergata, Via della Ricerca Scientifica 1, 00133 Rome, Italy

**Keywords:** circulating tumor cells, metastasis, CTC prognostic and predictive value, liquid biopsy

## Abstract

Circulating tumor cells (CTCs) currently represent a revolutionary tool offering unique insights for the evaluation of cancer progression, metastasis, and response to therapies. Indeed, CTCs, upon detachment from primary tumors, enter the bloodstream and acquire a great potential for their use for personalized cancer management. In this review, we describe the current understanding of and advances in the clinical employment of CTCs. Although considered rare and fleeting, CTCs are now recognized as key players favoring the development of cancer metastasis and disease recurrence, particularly in malignant melanoma, lung, breast, and colorectal cancer patients. To date, the advancements in technology and the development of several successful approaches, also including immunomagnetic enrichment allow for a reliable and reproducible detection and characterization of CTCs. Those innovative methodologies improved the isolation, quantification, and characterization of CTCs from the blood of cancer patients, providing extremely useful evidence and new insights into the nature of the tumor, its epithelial/mesenchymal profile, and its potential resistance to therapy. In fact, in addition to their prognostic and predictive value, CTCs could serve as a valuable instrument for real-time monitoring of treatment response and disease recurrence, facilitating timely interventions and thus improving patient outcomes. However, despite their potential, several challenges hinder the widespread clinical utility of CTCs: (i) CTCs’ rarity and heterogeneity pose technical limitations in isolation and characterization, as well as significant hurdles in their clinical implementation; (ii) it is mandatory to standardize CTC detection methods, optimize the sample processing techniques, and integrate them with existing diagnostic modalities; and (iii) the need for the development of new techniques, such as single-cell analysis platforms, to enhance the sensitivity and specificity of CTC detection, thereby facilitating their integration into routine clinical practice. In conclusion, CTCs represent a potential extraordinary tool in cancer diagnostics and therapeutics, offering unprecedented opportunities for personalized medicine and precision oncology. Moreover, their ability to provide *real-time insights* into tumor biology, treatment response, and disease progression underlines a great potential for their clinical application to improve patients’ outcomes and advance our understanding of cancer biology.

## 1. Introduction

Research on different biomarkers in different body fluids or rather circulating markers, the so-called “liquid biopsy” for the evaluation of various solid malignancies, has steadily increased [1].

In addition to the classic circulating biomarkers useful for cancer patients’ follow-up and diagnosis, such as the carcinoembryonic antigen (CEA) or the prostate-specific antigen (PSA), whose importance in routine oncology is well established, the liquid biopsy (LB) approaches include the detection of circulating tumor cells (CTCs), circulating tumor DNA (ctDNA), circulating miRNAs and tumor-derived extracellular vesicles (EVs). These components are released from the primary tumor cells and diffuse in the bloodstream promoting the formation of metastases in distant sites [2,3].

LB is defined as a non-invasive test representing a “surrogate diagnostic tissue” for molecular analysis. Its great potential resides in the fact that it is a minimally invasive tool, easy to perform, that can be repeatedly applied to the same individual, both in the diagnostic phase of cancer and during the follow-up. The LB approach will provide real-time information on the evolution of the individual patient’s disease, facilitating patient monitoring and response to treatment [4,5,6]. Furthermore, LB could be applied also to the analysis of both primary and metastatic (or micro-metastatic) sites providing a more heterogeneous picture of the entire tumor cell population compared to tissue biopsy samples.

The analysis of CTCs is currently a growing and promising area of research, in almost all types of solid tumors. Functional studies, mainly based on CTC-derived cell lines and CTC-derived explants, have demonstrated the ability to accurately depict the metastatic process. CTCs, in particular ctDNA or miRNA, show some additional advantages in clinical applications. They can be morphologically identified allowing for contextual genetic and molecular characterization through the identification of specific DNA driver mutations and tumor biomarkers [7,8]. CTC analysis currently presents a promising field to deepen our knowledge of the biology of the metastatic cascade and the mechanisms leading to the resistance to standard therapies, in both advanced- and even early-stage patients. Recently, some evidence also showed a significant potential of CTCs for early cancer diagnosis and screening, even though, at such an early stage of the disease, it seems very difficult to develop methods sufficiently sensitive for the isolation and characterization of CTCs [9].

The use of LB for the investigation of predictive biomarkers to assess in real time the molecular changes occurring in the tumor mass as well as the patients’ response to therapy represents one of the most powerful approaches in clinical oncology.

In the present review, we will illustrate the most recent knowledge and clinical applications of CTCs.

## 2. Metastatization Process

In total, 90% of cancer patients’ cancer-related deaths are related to the development of metastases. Metastasis formation is a multifactorial process resulting from the detachment of cancer cells from the primary tumor mass, the degradation of the extracellular matrix (ECM), and eventually their entrance into the bloodstream [10]. Initiation of metastasis is no longer considered merely a cellular autonomous event, or rather a linear cascade of events, but a process strongly influenced by the complex interaction of the tumor mass with the tissue microenvironments alongside a series of simultaneous, partially overlapping processes, leading to cells able to successfully metastasize due to the acquisition of new phenotypes while stripping old behaviors [11,12,13].

Two distinct interpretive models try to explain the metastatic cascade [12,14,15,16]. In the first one, at the early stage of the disease, cancer cells detach from the primary tumor to reach the lymph nodes or bloodstream. Tumor cells disseminating to regional lymph nodes are able to proliferate leading to the formation of solid metastases, whilst cancer cells that have entered into the bloodstream die or remain dormant. Subsequently, as the disease progresses, cancer cells present in lymph node metastases can spread to other organs through the bloodstream and form distant metastases. In this model, it has been hypothesized that the cancer cells initially unable to survive in the circulatory stream acquire the ability to form distant metastases through a selection process within lymph nodes. Thus, according to this model, the formation of distant metastasis is strictly dependent on cancer cell’s colonization of lymph nodes [16,17].

In the second model, postulated to explain the metastatic cascade, a genetic rewiring allows cancer cells to enter the bloodstream with high frequency and thus reach distant organs forming secondary metastases [17,18,19]. In this context, the passage through regional lymph nodes does not seem to be necessary. Specifically, this model could explain the distant metastases found in patients without malignant proliferation in the lymph nodes.

In both models, a second phase of blood dissemination may then occur from any metastases, formed at the level of distant organs or regional lymph nodes. This theory thus well fits the observed recovery of cytokeratin-positive cells in the peripheral blood (PB) of some breast cancer patients, even many years after primary tumor surgery [20].

Of note, CTCs that may secondarily spread, seem to be even more aggressive compared to those originating from first metastases, probably following the already occurring immuno-selection and the gaining of survival advantages into the bloodstream and into new target organs [21]. However, distant tissue colonization by CTCs has proven to be a mostly inefficient process. The recovery of a high number of CTCs (>1000 CTCs/mL of blood plasma) has been found with respect to a disproportionately low number of clinically detectable metastases [22], confirming that the spread of tumor cells in PB can occur early, but is usually detected late. At present, these proposed metastatic cascade models need further confirmation by the clinical setting and by gene characterization studies; nevertheless, both recognize CTC spreading as responsible for the process of metastatic dissemination.

## 3. Circulating Tumor Cells (CTCs): An Overview

### 3.1. CTCs as Single Cells

The first description of CTCs dates back to 1869 [2], whilst the first finding demonstrating the prognostic relevance of CTCs was published in the early 2000s. In this manuscript, the authors demonstrated that in patients with metastatic breast cancer, the enumeration of CTCs was a predictor of progression-free and overall survival [3]. Moreover, the CTC prognostic enumeration value was independent of the therapy followed by the patients (first or second line). Of note, CTC enumeration showed a major value with respect to “conventional” prognostic indicators such as tumor size, degree of differentiation, metastatic site, type of therapy, and the interval until the recurrence following the completion of primary surgery. Accumulating evidence suggests that the epithelial-to-mesenchymal transition (EMT), the main mechanism underlying cancer cell spreading, acts to promote CTC release from the primary tumor mass, eventually driving the development of distant metastases. For this reason, CTCs are considered a useful and formidable tool both for cancer diagnosis and follow-up considering their ability to replicate real-time tumor heterogeneity and the clonal evolution of cancer [23,24,25,26].

The activation of EMT promotes the acquisition of a completely different phenotype in CTCs. These changes consist of a reduction in the expression of epithelial markers, such as E-cadherin, occludins, and claudins, along with the gaining of mesenchymal hallmarks, including N-cadherin, vimentin, and fibronectin, conferring to cancer cells an increased aggressiveness. The disruption of cell–cell junctions (desmosome, tight, and adheren junctions) increased cancer cell migration and invasion [27,28,29,30]. Mesenchymal tumor cells pass inside the blood vessels and then escape through basement membranes, interstitial spaces, and endothelial barriers.

A key feature, common to both the CTCs’ intravasation and extravasation processes, is the transendothelial migration (TEM). During TEM, tumor cells, characterized by an EMT profile, cross the vascular endothelial layer. In physiological conditions, endothelial cells act as a barrier, blocking the movement of cells into or out of the blood. Mesenchymal cells, on the other hand, are able to cross the vascular endothelial layer during both the intravasation [31] and extravasation process [32]. This suggests that, in addition to the acquisition of significant metastatic potential by cancer cells, the compromise of vascular integrity at primary and metastatic sites also plays a functional role in facilitating TEM [33].

### 3.2. CTC Clusters or Microemboli

However, CTCs do not exclusively consist of single cells that dissociate from the primary tumor; rather, in most cases, they are represented by clusters of cells. In some cases, they are even referred to as microemboli, consisting of groups of CTCs, immune system cells, and platelets [34]. Of note, the behavior and features of individual CTCs are completely different from those of CTC clusters. The formation of CTC clusters requires the acquisition of a partial EMT phenotype by the cancer cells which allows them to increase their migratory capabilities while simultaneously maintaining the cell–cell interactions and thus an epithelial cell profile [23,35,36,37]. The collective migration requires stable cell-to-cell adhesion and a coordinated multicellular movement. This is the reason why the CTC clusters could include different cellular morphologies, i.e., both epithelial and mesenchymal. It has also been hypothesized that the existence of a leader mesenchymal cell drives tumor cells through the surrounding tissue. In contrast, single CTCs must weaken or completely lose their adhesive bonds with neighboring cells in order to infiltrate [36,38,39].

The importance of “collective cell migration” has recently been highlighted attributing invasive capacity to the entire multicellular aggregate or cluster. It has been suggested that the formation of microemboli confers to cancer cells benefits in terms of survival [40] and proliferation [41], facilitating the formation of distant metastatic lesions [42]. Indeed, it has been demonstrated that CTC clusters could give rise to metastases without extravasation, adhering directly to the vessel walls of arterioles and capillaries and proliferating within the vascular system [43,44,45]. Finally, only after the rupture of the capillaries’ walls do they lead to the formation of micro- or macro-metastases. This plasticity indicates that phenotypic transformation is not always required for tumor cells to acquire a higher motility potential. Therefore, given that not all metastatic tumor cells lose their epithelial features following EMT, it is important to consider the entire heterogeneous population of CTCs, including those with epithelial properties, in order to identify the true cells responsible for initiating metastasis [46].

### 3.3. CTCs and EMT

CTCs, both as single cells and clusters, that survive in circulation can become capable of exiting the bloodstream and colonizing distant organs forming metastatic tumors or could remain in these organs in a dormant phase [15,47,48]. In several carcinomas, it has been shown that the forced activation of the EMT promotes in these cells the formation of *filopodium-like* protrusions, which in turn induced CTC extravasation, proliferation, and eventually the production of metastases [49].

The EMT cascade, and the principle of the hybrid EMT phenotype, does not only regulate the metastatic process, but it also affects the CTC enrichment approaches. Indeed, the currently available CTC isolation techniques are mainly based on exploiting their epithelial nature used for their isolation of the expression of the epithelial marker EpCAM (epithelial cell adhesion molecule, CD326) [50,51]. However, it must be considered that the exclusive use of EpCAM as a biomarker to purify CTCs could lead to the loss of those subpopulations of CTCs that do not express the epithelial markers but only express mesenchymal markers, unless the simultaneous use of both epithelial and mesenchymal-related markers occurs. Thus, a combination of different epithelial and mesenchymal biomarkers could represent a practical solution to thoroughly characterize the heterogeneous CTC subpopulations and enhance their prognostic significance [30]. Figure 1 provides a graphical representation of the route followed by both single CTCs and CTC clusters in the formation of distant metastases [52].

## 4. Clinical Value of CTC Analysis

### 4.1. Different Approaches for CTC Isolation and Enumeration

CTCs, as previously mentioned, originate from different cancer lesions and include heterogeneous cell subpopulations. Of note, both the tumor microenvironment and any potential chemotherapy treatment contribute to their phenotypic and molecular characteristics. For all these reasons, the enumeration and subsequent molecular analysis of CTCs in clinical oncology is considered a powerful tool for studying tumor heterogeneity fundamental to the development of personalized oncology medicine [53,54,55].

Enrichment procedures are necessary due to the low number of CTCs found in the blood of patients, including those with metastatic disease (generally less than 10 cells/mL), and their identification and isolation have posed quite a few technical issues [53]. Defined as “rare events”, their purification and further molecular characterization are needed pre-analytical enrichment and separation processes. The selection mechanisms of CTCs in relation to their ability to invade and metastasize are still the subject of numerous studies. CTC enrichment techniques aim to separate them from blood cells by leveraging some of their morphological characteristics: (a) a different density in comparison to the blood resident cells, through the use of liquid gradient density kits [56,57]; (b) different cell sizes in relation to blood cells, allowing for their purification with the aid of specific porous filter membranes (the diameter size of CTCs ranges between 20 and 30 μm) [58,59,60,61]; and (c) immunomagnetic enrichment, which allows for the isolation of CTCs on the basis of the superficial markers expressed by the cells [62,63]. In this last approach, using specific markers to detect CTCs presents another challenge: CTCs express different markers depending on the tumor from which they have detached [54]. Most cancers are of epithelial origin, and thus the “universal” epithelial molecular marker EpCAM is used for CTC detection. However, different tumors exhibit different expression levels for EpCAM. For example, it has been demonstrated that the use of EpCAMs as biomarkers for CTC enrichment has yielded the best results in breast and prostate cancers, tumors characterized by a high level of epithelial marker expression [64].

In this context, the CellSearch^®^ platform (Menarini Silicon Biosystems, Florence, Italy) was the first system developed for CTC detection and enumeration [65]. The FDA approval received in 2004 has allowed for its widespread use for diagnostic purposes across many different types of tumors. It is a highly reliable and semi-automated approach able to detect and count CTCs in 7.5 mL of PB in metastatic cancer patients, and in 22.5 mL of PB in early-stage cancer patients [66]. The immunomagnetic enrichment operated by the CellSearch^®^ system mainly exploits the expression of EpCAM on the CTCs’ surface and is dependent on several key factors such as the number of cells per volume of blood drawn and the level of surface antigen expression of cells immersed in the magnetic field. It takes advantage of the lack of expression of epithelial markers from the peripheral blood (PB) cells because of their hematopoietic–mesenchymal origin [67]. Thus, the discrimination between CTCs originating from epithelial tumors and blood cells is easily possible. However, to avoid leukocyte contamination, an anti-CD45 antibody (common “*pan*” leukocyte antigen) is utilized to exclude white blood cells from the count [68]. Overall, the detection of CTCs is performed on a negative selection excluding all the CD45^+^ cells and on a positive selection isolating CTCs based on the expression of epithelial markers, such as EpCAMs, or different cytokeratins (CKs) such as CK7, CK8, etc. [69]. Several studies showed that CTCs could be detected in 20–54% of early-stage breast tumor patients by using CK19-based PCR assays [70]. In detail, the authors showed that the enumeration from two to five CTCs per 7.5 mL of blood identified a subgroup of patients with reduced survival. Although the loss of expression of epithelial markers may cause false negatives, excluding from the enrichment procedure CTCs exhibiting a full mesenchymal phenotype, CKs, and especially EpCAMs, remain the most commonly utilized marker for identifying CTCs. Of note, both EpCAMs and CKs are not only expressed by epithelial tumors but also by dendritic and normal epithelial cells. In addition, during malignant progression and the occurrence of EMT, these antigens are down-regulated, adding further difficulties to the detection of CTCs [71,72,73].

In light of these observations, it would be useful to use a panel of markers to collect as many CTCs as possible, in order to take into account both their original heterogeneity and the molecular changes occurring during disease progression. Recently, it has been reported that the endothelial marker CD146 could be useful for the detection of EpCAM^−^ CTCs. Of note, CD146-based selection of CTCs demonstrated the same clinical relevance^+^ of the EpCAM-based selection in poor-prognosis patients [74,75].

### 4.2. Clinical Application of CTC Enumeration in Advanced Cancer Patients: OS and DFS Prediction

It has been shown that CTC detection may correlate with worse outcomes in both progression-free survival (PFS) and overall survival (OS), regardless of lymph node status and adjuvant therapy [70]. One of the first indications of the prognostic potential of CTCs was reported by Cristofanilli and colleagues in 2004, in the *New England Journal of Medicine*. The authors reported that 60 to 70% of metastatic breast cancer patients were detected with ≥5 CTCs per 7.5 mL of PB, whereas in healthy control subjects, CTCs were rarely observed. In another retrospective study [76], Cristofanilli performed a pooled analysis of data collected from 18 different cohorts of metastatic breast cancer patients. The results obtained showed that, on the basis of the CTC count, MBC patients could be stratified as “Stage-IV aggressive” or “Stage-IV indolent” MBC patients. In particular, Stage-IV aggressive MBC patients were characterized by ≥5 CTCs/7.5 mL of PB in comparison to “Stage-IV indolent” patients in which <5 CTCs/7.5 mL of PB were detected, also characterized by a longer median OS [77].

To explore the potential use of CTC enumeration as a circulating biomarker for risk stratification in MBC patients, a study was conducted analyzing data obtained from 469 patients. The findings collected were organized in order to group patients sharing similar CTC “paths” during chemotherapy. This investigation led to the identification of four novel prognostic groups including a basal group formed by 50% of patients characterized by a better prognosis in comparison to the other three groups. These last three groups included 30%, 15%, and 5% of patients who share increasing amounts of CTC trajectories and are correlated with a higher risk of progression and death [78]. This novel “prognostic classification approach” supports the idea that the detection of CTCs could be used to introduce a novel staging classification of advanced disease [78]. A further confirmation of the successful use of CTC enumeration as a biomarker for cancer patients’ OS and DFS came from the clinical trial SUCCESS A, in which the CTC counts in 1087 early-stage breast cancer patients were assessed, before and two years after chemotherapy, through the CellSearch^®^ system. The results of the trial demonstrated that the presence of CTCs in the PB of patients two years after chemotherapy was predictive of a decreased OS (overall survival) and DFS (disease-free survival) [51,79]. Furthermore, the detection of CTCs, either before or after chemotherapy, was correlated to the insurgence of multiple-site or bone-only first distant disease. These clinical indications hold massive value, allowing clinicians to increase patient monitoring or guiding oncologists toward more targeted therapies [51,79]. In metastatic “non-small-cell lung cancer” (NSCLC), clinical data from 550 patients in association with the prognostic OS were obtained. The detection of 5 CTC/7.5 mL PB was associated with reduced PFS and OS [80]. In sixteen advanced NSCLC patients, the enumeration value of EpCAM-positive CTCs was evaluated as a prognostic biomarker for the OS. The presence of a high number of EpCAM-positive CTCs resulted in being associated with poor OS, while the presence of a low number of EpCAM-positive CTCs did not [51,81].

### 4.3. Clinical Application of CTC Enumeration in Early-Stage Cancer Patients

The predictive value of CTCs in indicating the potential recurrence of the disease has also been demonstrated in patients with early-stage colorectal carcinoma undergoing curative surgical resection. The identification of CK20^+^CEA^+^ CTCs within 24 h of blood-draw, is indicative of recurrence [51,82]. In a different study, focused on pancreatic ductal adenocarcinoma (PDAC), thirty-six patients were enrolled, and CTCs were detected through a centrifugation size-based approach, the CD-PRIME™ platform. Afterward, isolated CTCs were characterized and enumerated upon immunofluorescent labeling of the epithelial markers EpCAM and pan-CK (CK 8, 18, and 19). CTCs were detected in twelve patients (33.3%), and in these patients, a significantly higher rate of both early and systemic recurrence was found. The data obtained further support the possible clinical application of CTC enumeration in also predicting cancer recurrence at the early stages of the disease [51,83]. Recently, a consensus statement has been elaborated on regarding the assessment of circulating biomarkers also in metastatic prostate cancer (PCa) and CTC detection, and characterization has been introduced for clinical monitoring of cancer patients, a fundamental tool assuming both prognostic and predictive clinical value [84]. Indeed, it has been demonstrated that the increase in CTC counts significantly correlated with PCa progression: in 511 PCa patients, the rising CTC values following the first 12 weeks of chemotherapy were associated with worse OS in patients, independently from the type of treatment adopted [85].

Thus, in breast cancer patients, as in those with early-stage colorectal carcinoma and in PDAC patients, CTC detection and count should be considered an eligible approach to detect possible recurrence of the disease in advance.

In contrast, studies on localized prostate neoplasia did not show this correlation: research of CTCs, either by the CellSearch^®^ system or by specific transcript type evidence, rarely documented CTC isolation in patients with localized disease [86,87,88]. Similarly, also in non-small-cell lung cancer (NSCLC), the predictive value of CTCs has not been demonstrated. However, Chemi and colleagues, in 2019, enrolled 100 patients with early-stage NSCLC and performed CTC isolation in blood specimens collected from the pulmonary vein during surgery. They found that 48% of patients were positive for CTCs. Intriguingly, a high number of CTCs (≥5 CTCs/mL) detected in patients before radiation treatment, as well as the presence of CTCs post-therapy, positively correlate with an increased risk of distant recurrence [89]. The results obtained in this study strongly suggest that CTC enumeration could be a useful tool for also monitoring cancer recurrence in NSCLC patients. In hepatocellular carcinoma, preliminary evidence shows that the enumeration and molecular characterization of CTCs could also be clinically applied for the early diagnosis and staging of cancer [90,91].

This additional potential use of CTCs in patients with early stages of various cancers could potentially lead to more personalized and specific therapies for the patient. Thus, the detection and characterization of CTCs to be used as indicators of minimal residual disease represents the future goal to achieve. Though, to date, the monitoring of patients with metastatic disease represents the main practical application of CTCs analysis.

## 5. Clinical Significance of the Molecular Characterization of CTCs

Together with CTC enumeration, another promising approach is the clinical evaluation of the molecular signature of CTCs. Indeed, CTC molecular profiling could allow for the assessment of prognosis in both early-stage and advanced disease, gaining the same clinical utility as other well-established tumor markers. Additionally, the characterization of the molecular profile of CTCs could strategically lead to the identification of specific therapeutic targets, resistance mechanisms, and possible responses to therapies, occurring at that specific moment in cancer patients, further contributing to the advancement of precision medicine in oncology [51].

To date, treatment decisions are based not only on clinical assessments but also on the consideration of the biological features of the primary tumor at the time of surgery, which are often redefined even many years after onset [48]. However, it is well known that the biology of the metastatic disease may be different from that of the primary disease. Molecular characterization of these cells in numerous perspectives studies has highlighted a high level of heterogeneity in the expression of genes coding for growth factors receptors, proteases, adhesion molecules, and histocompatibility antigens. Specifically, CTCs could be molecularly characterized in order to depict their stemness potential and their possible resistance to traditional chemotherapies. Compared to protein profiling-based techniques, analyzing the expression profiles of multiple genes results in a more sensitive detection of tumor-specific molecular markers, offering greater assurance to clinicians in selecting more specific and potentially more effective targeted treatments. One of the main limitations is related to the stringent specificity of markers adopted for CTC selection [92]. Indeed, the use of erroneous CTC markers could lead to false positives or negatives and to the possible contamination of the CTC samples by non-tumor-specific derived nucleic acids. Moreover, the hazard of cell lysis could preclude any possible subsequent analysis of CTCs. To overcome these constraints, different studies focused on the development of reliable experimental protocols for the molecular characterization of CTCs. Up to now, the results obtained lay the basis for the successful development of this clinical strategy in order to improve cancer patients’ benefits.

CTCs, besides reflecting the molecular features of the cancer cells composing the primary tumor mass, can also undergo additional mutational events that result in additional genetic mutations, different from those present at the onset. These new mutations could reveal novel mechanisms of drug resistance adopted by the tumor cells, as well as provide information about the acquisition of a higher metastatic potential by the tumor [93]. The definition of the molecular heterogeneity of CTCs, occurring at onset and during disease progression, has been demonstrated to be a useful tool for the specific characterization of several types of cancer, i.e., colorectal and prostate cancer [94,95].

Recently, cancer treatment regimens and their perspectives have been completely transformed and enriched by the introduction of immunotherapy. In particular, therapies targeting the immune checkpoint inhibitors programmed cell death protein-1/programmed cell death-ligand 1 (PD-1/PD-L1) interaction positively influenced the outcome of various forms of cancer such as NSCLC and malignant melanoma. The analyses of the PD-1/PD-L1 immune checkpoints in the CTC isolate from the PB of patients undergoing immunotherapy could also be used as a circulating biomarker for the prediction of diagnosis and treatment efficacy for patients receiving immunosuppressive drugs [96,97,98,99,100]. However, it has been shown that the predictive value of the PD-1/PD-L1 detection in CTCs varies depending on the tumor stages. Following the data and subsequent validation obtained in lung cancer, breast cancer, and gastric cancer, as well as in melanoma, the molecular profile of the PD-1/PD-L1 immune checkpoint in CTCs could potentially be used as a standalone prognostic indicator for immunotherapy [96,97,98,99,100].

Another extremely interesting aspect concerns the identification of specific molecular markers on tissue samples predictive of response to therapy. This approach could result in a strong improvement in the outcome for early-stage patients following treatment. However, in clinical practice routine, tissue samples may not be available for this kind of analysis, due to depletion of material or not being adequate to represent metastatic disease arising several years after the primary tumor. CD45 depleted CTC-enriched fractions expressing CXCR4, and the alpha-chemokine receptor specific for stromal-derived-factor-1, isolated from the PB of prostate cancer patients through the CellSearch^®^ platform, documented poor prognosis in patients treated by radiotherapy [101]. Of note, radiotherapy strongly affected CTC count, causing their drastic decrease in twenty-four metastatic prostate cancer patients, while a CXCR4^+^ subpopulation of CTCs still persists up to three months upon treatment, suggesting their resistance to local radiotherapy [84,85,102].

In a recent study including a cohort of 1220 breast cancer patients staged I-III, with more than a ten-year follow-up, it has been found that the presence of CK-19^+^ CTCs was prognostic for early relapse [103]. In 100 early breast cancer patients, the prognostic significance of the level of one of the main EMT transcription factors, TWIST1, was also investigated together with some stem cell transcripts, such as CD24, CD44, and ALDH1, in enriched EpCAM^+^ CTCs, obtaining promising results [51,103,104]. In metastatic breast cancer, the molecular profile of CTCs selected using specific EMT and stem cell markers, such as TWIST1, ESR1, PGR, HER2, EGFR, and CK-19, revealed that in the CD44^high^/CD24^low^ and in the ALDH1^high^/CD24^low^ CTC populations, the combined expression of the CK-19 gene with the HER2 overexpression was correlated with OS [105].

Strati and colleagues recently demonstrated that, in breast cancer patients (early-stage and metastatic patients), the CTC expression of epithelial markers such as CK19, EpCAM, and the bone marrow mammaglobin-1 (SCGB2A2), a specific breast cancer marker, was not correlated to the expression in CTCs of a panel including six genes indicative of the presence of CTCs (EMP2, SLC6A8, HJURP, MAL2, PPIC and CCNE2) [106,107].

All these findings supported the idea that the molecular signature of CTCs can be used to modify the staging system of advanced disease, providing additional information to the clinician, even if it may not fully predict patient outcomes, such as prognosis and specific therapies. The clinical relevance of CTCs for the development of personalized medicine in the field of cancer has been further corroborated by multiomics approaches [108]. The phenotypic, genomic, epigenomic, and transcriptomic characterization of CTC-derived cell lines isolated from the PB of a mice model of breast cancer revealed that a multiomic approach could be applied to drug testing in order to identify the best therapeutic approach for each patient [109].

Besides the acquisition of mutations by CTCs, it is also important to consider the occurrence in CTCs of epigenetic alteration. A recent study performed in CTCs isolated from CRC patients revealed a completely different methylome profile of a metastatic competent CTC-derived cell line in comparison to commercially available primary CRC cell lines [110]. Likely, the deciphering of the methylome profile of CTCs isolated from lung cancer patients uncovered a unique DNA methylation signature of CTCs, which was entirely distinct from that observed in the primary cancer tissues [111]. In particular, the authors found that the isolated CTCs were characterized by a mixed epithelial/mesenchymal (E/M) phenotype: DNA hypomethylation in the promoter of the EMT transcription factors *TWIST1*, the mesenchymal *CDH2* and the stemness-related genes *ALDH1A1* and *CD44*. In contrast, the authors found a hypermethylated state of the promoters of the EMT transcription factors *Snai2* and the mesenchymal gene *Vimentin* [111]. In metastatic breast cancer, the methylation of the estrogen receptor1 (*ESR1*9), which encodes estrogen receptor alpha (ERα), has been associated with patients’ resistance to the everolimus/exemestane therapeutic approach [112]. Notably, in breast cancer patients, the modulation of CTCs’ methylome has also been described in relation to CTC clusters. Gkountela and collaborators described a specific methylation profile of CTC clusters in comparison to single CTCs. CTC clusters showed hypomethylation in transcription factors of stemness genes (*OCT4*, *NANOG*, *SOX2*, and *SIN3A*), and their dissociation induced by a Na^+^/K^+^ ATPase inhibitor led to the rewiring of DNA methylation sites and reduced their metastatic potential [42]. The role of epigenetic modulation and its clinical relevance in CTCs has been extensively reviewed by Vasantharajan and colleagues [113].

## 6. Relevance of Circulating Stem Cells

Another extremely complex aspect, worthy of a standalone review, concerns the circulating stem cells (CSCs). CSCs are defined as a small subset of cells with “stem-like” features similar to the normal tissue stem cells. They are characterized by both a self-renewal and differentiation capacity and, most importantly, by the ability to initiate tumorigenesis [114,115]. The main role of CSCs seems to be the repopulation of the cell subpopulation residing in the primary tumor. Therefore, the possible targeting of these CSCs could lead to the eradication of cancer resulting in an effective cure [116,117,118]. The first clear evidence of the existence in the PB of CSCs as a subset of circulating stem tumor cells (CSTCs) has been documented in acute myeloid leukemia (AML). Their detection has been obtained through antigens normally used to identify healthy hematopoietic stem cells. The identification of CSTCs in AML fueled new research into other types of tumors leading to the demonstration of the existence of CSTCs even in solid tumors [119].

Therapeutic resistance, recurrence of the underlying tumor, and lack of curative treatments in metastatic disease raise questions about the real efficacy of conventional cancer therapies, as well as whether these therapies target the “right” cells. In fact, these therapeutic treatments could not affect CSTC proliferation and survival considering their resistance to many treatments, including chemotherapy and radiotherapy [120]. CSTCs are able to escape antiproliferative chemotherapy probably due to their relative dormancy [121,122,123,124]. Surviving standard cancer therapies allows them to potentially regenerate the tumor even after most cancer cells have been killed [125]. In breast cancer patients, CSTCs 5 to 10 or even more years after surgery have been detected. Indeed, 20% of women develop disease progression in the period from 7 to 25 years after radical mastectomy [126]. In addition, our research group has also demonstrated the presence of circulating melanoma cells (CMCs) up to 18 years after the onset of the disease [127].

Recently, an interesting study performed in breast cancer demonstrated the simultaneous presence of cancer stem cells and cancer cells undergoing EMT [128]. However, in a prospective study including 22 patients with nonspecific-type invasive breast carcinoma, through flow cytometry have been identified in the PB of cancer patients several cells’ subpopulations [129]. Specifically, PB samples were collected before or after biopsy and after minor surgical tumor removal (biopsy) in the absence of neoadjuvant chemotherapy. The authors found that the minor surgical removal of the tumor increased the release of two different CTC subpopulations: (1) Epcam^+^CD45^−^CD44^−^CD24^−^Ncadh^−^ CTCs, which lack both stemness and EMT features, and (2) Epcam^+^CD45-CD44^+^CD24^−^Ncadh^−^, indicative of staminal CTCs, in patients’ PB. Conversely, EpCAM^−^ CTCs were strongly reduced after surgery. These intriguing findings suggest that the CTC subpopulation characterized by the presence of stemness markers, known as CSTCs, likely possesses a higher metastatic potential than other circulating tumor cells. This grants them the ability to disseminate and initiate the metastatic process, contributing to disease recurrence even after seemingly successful treatment [129]. Three years later, the same authors deepened their investigation by analyzing a cohort of 135 patients with invasive breast carcinoma. They performed a single-cell immunofluorescence analysis on the patients’ tumor tissues in order to identify stemness and EMT signs in single tumor cells (STCs). Besides the evidence of intra- and inter-tumor heterogeneity in STCs, they found that the presence of STCs with an epithelial profile correlated with a higher risk of developing metastases [130].

Our group focused the attention on patients with primary melanoma and/or metastatic disease, including patients classified at least as AJCC staged pT1B (a transition from the radial to vertical growth phase). The approach used for the isolation and molecular characterization of melanoma CTCs (CMCs) combines an immunomagnetic enrichment procedure with a negative selection for the common leukocyte antigen CD45, followed by positive selection targeting the CD146 antigen (expressed in more than 80% of metastatic melanomas) and ABCB5 melanocyte stem antigen [75,131,132]. Such enrichment allowed us to identify circulating neoplastic cells with a stem–mesenchymal phenotypic “signature” (CD45^−^CD146^+^ABCB5^+^) whose gene expression analysis provides fundamental information about the biology of the neoplasm [133,134]. Effectively, this combined approach greatly increased the number of isolated CMCs, allowing for a comprehensive real-time characterization of the cancer.

During the metastatic cascade, the EMT is aberrantly activated, allowing CTCs to migrate, translocate into the bloodstream, and eventually reach distant sites. This process often occurs during a specific stage (invasion and/or intravasation) of tumor progression. We analyzed the expression levels of two specific gene panels. The first included genes that are specific for the characterization of metastatic cell–cell adhesion and invasion phenotypes (*MCAM/MUC18/CD146*, *CDH5* (coding for VE-Cadherin), *CDH1* (coding for E-cadherin)*, CDH2* (coding for N-cadherin)*, MMP9*, *MMP2*, *VEGF*, and *bFGF*), while the second panel encompassed the main EMT-related gene, such as *HFN1*, *CD146/MCAM*, *CDH1*, *CDH2* and *Vimentin*, and the EMT transcription factors (EMT-TF) *TWIST1*, *SNAI2* and *ZEB1*. We molecularly characterized melanoma patients at onset and during follow-up, profiling a metastatic expression phenotype capable of defining the minimal residual disease of each patient under investigation, regardless of the presence or absence of somatic driver mutations at disease onset [135]. We developed a protocol with high sensitivity, able to “mimic”, as closely as possible, the circulating minimal residual disease condition which consequently allows for the detection of this rare subpopulation of CMCs with specific stem–mesenchymal phenotypic “signature” (CD45^−^CD146^+^ABCB5^+^). Quantitative expression panels performed at disease onset and/or during clinical treatments (targeted therapy or checkpoint-inhibitor therapy), showed that therapies could strongly affect the EMT. Indeed, our data suggested a significant decrease in the EMT-related genes in the CMCs isolated from the PB of patients with a clinical remission *status.* Conversely, in patients experiencing disease progression or refractoriness to therapy, we observed the persistence of EMT genes in the detected CMCs [133,134,135]. A schematic diagram of our current method for the detection, isolation, and enrichment of actively metastasizing CMCs, is reported in Figure 2. Additionally, in Table 1, we reported all the studies thoroughly revised in this manuscript.

## 7. Perspectives and Conclusions

Currently, the prognostic relevance of CTC enrichment is well established in many types of solid cancers. Indeed, CTC enumeration, during the patient’s treatment, could be an additional, non-invasive, and easily applicable clinical tool to predict the patient’s response to therapy. An additional approach, capable of providing even more detailed information to clinicians, involves the molecular characterization of CTCs, in addition to their enumeration. In fact, the molecular profile of CTCs holds the potential to track real-time molecular changes occurring during cancer progression, giving clinicians the opportunity to counteract cancer spread at very early stages. Moreover, the molecular characterization of CTCs could reveal the presence of resistance mechanisms acquired by tumor cells, guiding oncologists toward different targeted therapies. It must be considered that the molecular characteristics of CTCs evolve as tumor foci progress and during tumor progression. CTCs can aggregate, forming groups or clusters with differentiation profiles and acquiring additional mutations that are related to the emergence of new tumor subclones and that fuel tumor heterogeneity. Detection of CTC clusters seems to be strictly related to the increased metastatic potential of cancer cells. This finding suggests the targeting of CTC clusters could have a stronger impact on cancer progression in comparison to therapies aiming for a reduction in single CTCs. Interestingly, CTC clusters had been recently found in early colorectal cancer stages and not only in advanced stages of the disease [137]. As a result, CTCs partially reflect the spectrum of tumor mutations and their heterogeneity and can be considered as a snapshot of the developmental disease at that specific moment, although potentially not exhaustive. Their genetic and expression profiles may differ from those of the primary tumor cells, allowing for specific functional investigations over different periods of time or during follow-up. This approach could allow for the selection of the most suitable targeted therapy regimen according to genotypic changes in CTCs, and its efficacy could be evaluated over time.

Of note, many studies have demonstrated the clinical potential of liquid biopsy even in the early diagnosis of cancer, strongly improving patient prognosis and survival [1,2,8].

The clinical utility of CTC enumeration and molecular characterization, as shown so far, has led to the design of different clinical trials [72,77,139,140,141,142]. All those findings support the idea that the presence of CTCs together with their molecular signature could be used to modify the staging system of advanced disease, providing additional information to the clinician and significantly contributing to the development of personalized medicine [143].

Once the enormous clinical potential of CTCs was established, both in the early and advanced stages of cancer, researchers focused their efforts on developing reliable and sensitive protocols for CTC identification and molecular profiles, even at the single-cell level. However, these efforts have been met with the complexity of this system, which, day by day, reveals increasing heterogeneity in the CTC subpopulations depending on many factors, such as the stage of the disease, the tumor microenvironment, and especially the type of cancer being analyzed and the possible chemotherapeutic regimens.

To date, we would like to develop in other solid tumors such as lung, colon, and pancreatic cancers an enrichment protocol that, as for melanoma, could allow for the detection of the circulating minimal residual disease condition. We hypothesize to detect and enrich specific stem-mesenchymal CTCs subpopulations, which could be useful to provide additional information about possible evolution of neoplasia. It will be essential to accurately identify the targetable antigens for the selection, identification, and enrichment of CTCs, with the aim to characterize perineural invasion in pancreatic cancer rather than angiogenesis, which is extremely involved in invasion and metastasis in lung cancer.

All of the data here reported underline the clinical value of CTCs for also generating in vitro and in vivo models to obtain an in-depth investigation of a wide panel of areas of cancer research, such as therapy, disease evolution, and real-time genomics characterization. The in vitro propagation of CTC-derived cell lines has been successfully established following CTC isolation from the PB of a metastatic breast cancer patient at different time points. The molecular profile of the isolated cultured CTCs revealed the deregulation of genes involved in drug resistance, xenobiotic and energy metabolism, stem cell properties, and plasticity. Zhang et al. were the first to establish primary cultures from CTCs obtained from patients with advanced breast cancer and to identify markers specific for brain metastases in a subpopulation of cultured cells [136]. Their findings suggested the potential use of this approach to investigate the metastatic potential of isolated CTCs. Likewise, Cayrefourcq and coworkers obtained a stable CTC line from a colorectal cancer patient showing tumorigenic and pro-angiogenic properties [144].

Although CTCs have been identified and studied in most malignancies, there is still a lack of solid knowledge about the biological characteristics of these cells and their life cycle. In particular, there is uncertainty regarding the timing of their first release into the bloodstream, their genetic profile in relation to the tumor, the mechanisms of intravasation and extravasation, and their means of survival in circulation. Their low frequency in the blood, heterogeneity, and poor survival also contribute to these challenges.

Despite the stimulating and encouraging evidence reported so far, important limitations are currently evident. The low frequency of such cells in patients’ blood, together with the complexity and cost of the procedure required for their isolation, have limited the large-scale dissemination in clinical practice of circulating tumor cell analysis. The comparison of data obtained through different technologies on the same type of samples showed low concordance. In this context, better methods for culturing and expanding CTCs are needed to study their molecular profile and to study the possible modulation of their behavior based on the pressure exerted by the local microenvironment. Possibly, the rapid improvement of technologies to detect, isolate, and characterize CTCs will conclusively demonstrate their clinical validity and utility in the future. Currently, there are technologies that isolate CTCs regardless of their phenotypic characteristics, so there are no specific protocols able to simultaneously isolate all the CTC subpopulation residents in the PB of patients. However, even with such methods, the number of CTCs will be too small to adequately characterize their aggressive and metastatic potential.

Standard cell lines for quality control of CTC analysis, inter-laboratory projects, and official setup are needed urgently. This would provide a high sensitivity and specificity to the system, capable of detecting early-stage tumors. Several organizations, encompassing the European Liquid Biopsy Society (ELBS), the International Liquid Biopsy Standardization Alliance (ILSA), the International Society of Liquid Biopsy (ISLB), the International Federation of Clinical Chemistry (IFCC), and the European Federation of Laboratory Medicine (EFLM) are working to establish consistent, sustainable, and sensitive CTC-based clinical tools to improve patients’ well-being.

In conclusion, CTCs represent a stimulating challenge to prompt a paradigm shift in cancer diagnostics and therapeutics, offering unprecedented opportunities for personalized medicine and precision oncology. Moreover, the ability of CTCs to provide real-time investigation of tumor biology, treatment response, and disease progression holds great potential for stable employment of CTC analysis in clinical practices improving patient outcomes and advancing our understanding of cancer biology. Moreover, the establishment of CTC-derived cell lines will represent a new opportunity to decipher the metastatic cascade and, hopefully, to find successful targeted therapies to counteract cancer spreading.

## Figures and Tables

**Figure 1 biomedicines-12-02137-f001:**
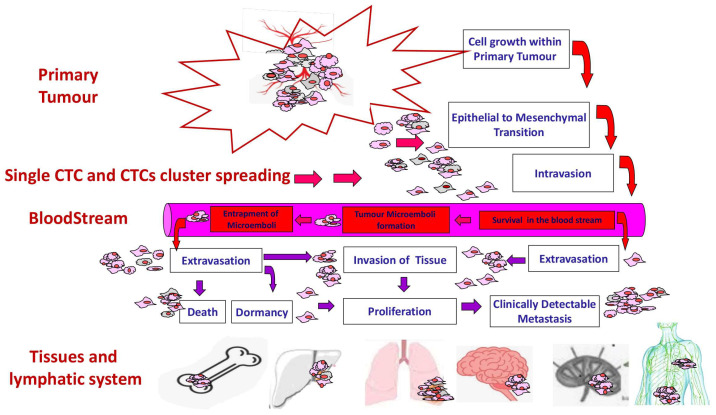
Path of metastasizing tumor. CTCs, as single cells or clusters, detache from the primary tumor, enter into the bloodstream and, upon extravasation, establish distant metastases; modified from Rapanotti et al., 2017 [52].

**Figure 2 biomedicines-12-02137-f002:**
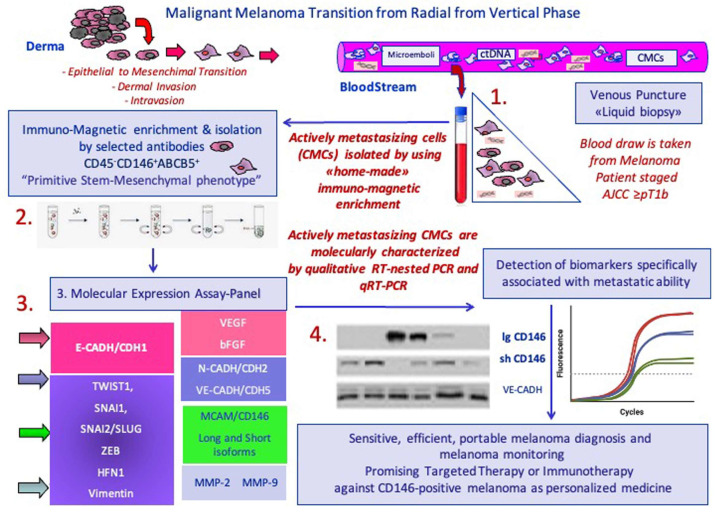
Schematic diagram of our current method for detection, isolation, and enrichment of actively metastasizing CMCs.

**Table 1 biomedicines-12-02137-t001:** Studies on CTC isolation protocols, prognostic significance, and CTC’s role reviewed in the manuscript.

Tumor	CTC Detection Technique	Patients Enrolled	Aim of the Research on CTCs	Clinical Application	Reference
Advanced melanoma	Multiparametric flow cytometry	40	Molecular characterization	Response to therapy	[96]
Melanoma	CD146^+^ABCB5^+^ immunomagnetic enrichment	30	CD146 expression	Monitoring of minimal residual disease	[75]
Advanced melanoma	CD146^+^ABCB5^+^ immunomagnetic enrichment	22	Molecular characterization	Correlation with disease progression	[133]
Advanced melanoma	CD146^+^ABCB5^+^ immunomagnetic enrichment	8	Molecular characterization	Minimal residual disease monitoring, for melanoma in high-/low-risk patients	[134]
Advanced melanoma	CD146^+^ABCB5^+^ immunomagnetic enrichment	7	Molecular characterization	Correlation with disease aggressiveness	[135]
Breast cancer	Magnetic activated cell sorting (MACS)	35	Molecular characterization	Response to treatment	[69]
Breast cancer	Immunomagnetic nanosphere (IMN)	164	Enumeration	Prognostic analysis	[70]
Breast cancer	Cytospins and immunofluorescence	198	Molecular characterization	Prognostic analysis	[98]
Breast cancer	RT-qPCR	36 metastatic 90 early stage	Molecular characterization	Prognostic analysis	[107]
Breast cancer	CMx (“Cells captured in Maximum”) platform	-	Multiomic approach	Drug screening	[109]
Breast cancer	Multiparametric flow cytometry	38	Molecular characterization	Prediction of brain metastases	[122]
Breast cancer	Immunomagnetic enrichment	36	Enumeration	Risk of recurrence	[125]
Breast cancer	Flow cytometry	22	Enumeration and molecular characterization	Role of surgery in the release of CTCs	[129]
Breast cancer	Immunomagnetic enrichment	8	Molecular characterization	Cancer diagnosis and aggressiveness	[136]
Colorectal cancer	Ficoll-Hypaque density cushion	64	Molecular characterization	Markers to identify malignant intestinal diseases	[82]
Colorectal cancer	EasySepTM (immunomagnetic bead isolation)	8	Molecular characterization	Target identification for personalized medicine	[94]
Colorectal cancer	CTC-derived cell line	1	Epigenomic characterization	Target identification for personalized medicine	[110]
Colorectal cancer	Veridex	31	Molecular characterization	Target identification	[123]
Breast cancer	Affymetrix HG-U133P microarray chips	9	Molecular characterization	Prediction of brain metastases	[137]
Breast cancer	Parsortix GEN3D6.5 cell separation cassette	43	Epigenetic characterization	Target identification for personalized medicine	[42]
Early breast cancer	Real-time monitoring during PCR	1220	Enumeration	Prognostic analysis	[103]
Early breast cancer	Immunomagnetic Ber-EP4-coated capture beads	100	Molecular characterization	Prognostic analysis	[104]
Early breast cancer	CellSearch^®^ system	1087	Enumeration	Prediction of distant metastases	[79]
Early-stage hepatocellular carcinoma	TSF platform	105	Enumeration	Prediction of recurrence	[91]
Early-stage non-small-cell lung cancer	CellSearch^®^ system	100	Enumeration and molecular characterization	Prediction of recurrence	[89]
Gastric cancer	Magnetic positive selection (beads)	70	Molecular characterization	Prognostic analysis	[97]
Hepatocellular carcinoma	CanPatrol™ CTC assay technology (size based)	176	Assessment of CC phenotype	Prognostic analysis	[90]
Lung cancer	Magnetic enrichment CTC-LMD extraction	15	Epigenomic characterization of E/M phenotype	Clinical diagnosis	[111]
Melanoma	RT-PCR	175	Molecular characterization	Prediction of clinical outcome	[132]
Metastatic breast cancer	CellSearch^®^ system	177	Enumeration	Prediction of survival	[76]
Metastatic breast cancer	Retrospective pooled analysis	2436	Enumeration	Staging of metastatic breast cancer	[77]
Metastatic breast cancer	Epigenomic characterization	122 74 early 48 metastatic	ESR1 methylation	Response to therapy	[112]
Metastatic breast cancer	CellSearch^®^ system	469	CTC-derived cell line	Drug testing	[138]
Metastatic breast cancer	Cytospins and immunofluorescence	130	Molecular characterization	Prediction of poor outcome	[105]
Metastatic breast cancer	Immunomagnetic Ber-EP4-coated capture beads	46	Molecular characterization	Prognostic analysis	[106]
Metastatic breast cancer	-	1994	Clinical trial Enumeration	Prediction of OS and PFS	[139]
Metastatic breast cancer	CellSearch^®^ system	1933	Clinical trial Enumeration	Prediction of clinical outcome	[140]
Metastatic prostate cancer	CellSearch^®^ system	276	Enumeration	Prediction of OS	[72]
Non-small-cell lung cancer	CellSearch^®^ system	550	Enumeration	Prognostic indication of OS and PFS	[80]
Non-small-cell lung cancer	ISET filters (size based)	106	Enumeration and molecular characterization	Response to therapy	[99]
Non-small-cell lung cancer	CellSearch^®^ system	127	Molecular characterization	Response to therapy	[100]
Pancreatic ductal adenocarcinoma	CD-PRIME™ platform (size based)	36	Enumeration	Indicators of early recurrence	[83]
Pancreatic cancer	Vimentin or EpCAM immobilized microfluidic chip	146	Clinical trial	Response to therapy	[141]
Prostate cancer	CellSearch^®^ system	131	Enumeration	Effect of radiotherapy on CTC release	[101]
Prostate cancer	CellSearch^®^ system	511	Enumeration	Disease progression	[85]
Prostate cancer	Automated fluorescence microscopy	47	Molecular characterization	Assessment of cancer aggressiveness	[95]
Prostate cancer	CellSearch^®^ system	24	Enumeration and molecular characterization	Prognostic analysis	[102]
Prostate cancer	GILUPI CellCollector^®^ CellSearch^®^ system	188	Enumeration	CTC identification	[86]
Prostate cancer	CTC-chip	55 36 localized 19 metastatic	Enumeration and molecular characterization	Identification of targets for therapy	[87]
Prostate cancer	CellSearch^®^ system	65	Enumeration	Response to therapy	[88]
Prostate cancer	Automated immunofluorescent staining	255	Clinical trial	Personalized medicine	[142]

## Data Availability

No new data were created or analyzed in this study.

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
