# Peer review of "Circulating Tumor Cells: Origin, Role, Current Applications, and Future Perspectives for Personalized Medicine"

_biomedicines, 2024, doi:10.3390/biomedicines12092137_

Round 1
Reviewer 1 Report
Comments and Suggestions for Authors
The authors summarized current knowledge in origination and the role of CTCs, their clinical applications and future perspectives of clinical utility in many types of cancer with focusing on melanomas.
I have several notes and recommendations:
According to instructions for authors “Review: Reviews offer a comprehensive analysis of the existing literature within a field of study, identifying current gaps or problems. They should be critical and constructive and provide recommendations for future research.”
From this point of view, own results of authors, which were obtained from patients with melanomas, were over-represented compared to other cancers. Moreover, the diagram was developed on the basis of three studies with only 21, 8 and 9 evaluated patients. To decrease this effect, CTC results from different cancers (Paragraphs 4 and 5) could be summarized in some tables of figures. Diagram at Figure 2 could be an example of specific strategy for melanoma diagnosis and monitoring.
· The verified fact that clusters of CTCs have higher metastatic potential was noticed Perspectives and Conclusion, but was not visualized at Figure 1. There is also the possibility that several-cell-CTC clusters can be detached from tumor mass and then these clusters differently intravasate into the blood vessels.
· Row 308 - …2 < CTC-counts > 5 out of 7.5 ml PB… ???
· Figure 2. – texts have a poor brightness and they are not readable.
For proportional representation of current knowledge, I recommend major revision.
Author Response
Reviewer 1
The authors summarized current knowledge in origination and the role of CTCs, their clinical applications and future perspectives of clinical utility in many types of cancer with focusing on melanomas. I have several notes and recommendations:
-According to instructions for authors “Review: Reviews offer a comprehensive analysis of the existing literature within a field of study, identifying current gaps or problems. They should be critical and constructive and provide recommendations for future research. ”From this point of view, own results of authors, which were obtained from patients with melanomas, were over-represented compared to other cancers. Moreover, the diagram was developed on the basis of three studies with only 21, 8 and 9 evaluated patients. To decrease this effect, CTC results from different cancers (Paragraphs 4 and 5) could be summarized in some tables of figures. Diagram at Figure 2 could be an example of specific strategy for melanoma diagnosis and monitoring.
We thank the reviewer for the relevant and appropriate suggestions. We agree with the reviewer. For this purpose, we have included as suggested and requested, a summary table with the data reported relating to the works cited by us. This certainly makes our manuscript more usable and agreeable. So we inserted in the main text, at the end of paragraph n°5 the Table 1” Studies on CTCs isolation protocols, prognostic significance and CTCs role reviewed in the manuscript.”
-The verified fact that clusters of CTCs have higher metastatic potential was noticed Perspectives and Conclusion, but was not visualized at Figure 1… There is also the possibility that several-cell-CTC clusters can be detached from tumor mass and then these clusters differently intravasate into the blood vessels.
We described briefly in paragraph 3, the important role of the CTC clusters, emphasizing what was actually ”missed” in the first draft. We have also modified figure 1, emphasizing CTC cluster and CtC withpresence.
Row 308 - …2 < CTC-counts > 5 out of 7.5 ml PB… ???
We apologize for the inconvenience. We have modified the text.
Figure 2. – texts have a poor brightness and they are not readable.
We modified both figures, Figure 1 and Figure 2, “acquiring" them with more effective programs to improve their definition. Hoping that this happened.
Reviewer 2 Report
Comments and Suggestions for Authors
While the review claims to address the current status of the issue of circulating tumor cells (CTCs), in my opinion, it fails to convey the heterogeneity of CTCs and neglects the contributions of certain scientific teams, such as Perelmuter et al. The clinical significance of these teams' work is also not disclosed, despite a substantial body of literature on the relationship between cancer stem cells (CSCs) and their molecular characteristics and cancer progression. In this regard, the review requires improvement.
The visual presentation of Figures 1 and 2 requires significant enhancement, please utilize specialized graphic tools to make them more aesthetically pleasing and easier to comprehend. Additionally, please correct the font layout in lines 102, 307, 308, and 375 to ensure consistency throughout the manuscript.
Comments on the Quality of English Languageno
Author Response
Reviewer 2
While the review claims to address the current status of the issue of circulating tumor cells (CTCs), in my opinion, it fails to convey the heterogeneity of CTCs and neglects the contributions of certain scientific teams, such as Perelmuter et al. The clinical significance of these teams' work is also not disclosed, despite a substantial body of literature on the relationship between cancer stem cells (CSCs) and their molecular characteristics and cancer progression. In this regard, the review requires improvement.
The reviewer raised an interesting question that we adopted. We regret having given the impression of neglecting such findings. It was not our intentions and, as we reported in the text we implemented the text accordingly with what was suggested. However, we think the topic “circulating stem cell,” (CSCs) deserves a review in its own right. Nonetheless, the suggestion to report some of the exciting results back to what was published by Perelmuteret al. actually helps to improve our manuscript. The publications on CSC are truly many and impressive, we have therefore dedicated ourselves to reporting those findings, the "closest" to the topics we have decided to describe, such as the antigenic and molecular characterization of CSC stemness and epithelial to mesenchymal transition. The additional publications have obviously been included in the text and in the References paragraph.
The visual presentation of Figures 1 and 2 requires significant enhancement, please utilize specialized graphic tools to make them more aesthetically pleasing and easier to comprehend.
We modified both figures, Figure 1 and Figure 2, “acquiring" them with more effective programs to improve their definition. Hoping that this happened.
Additionally, please correct the font layout in lines 102, 307, 308, and 375 to ensure consistency throughout the manuscript.
We apologize for the inconvenience. We have modified and unified the font of tables and paragraphs in the main text.
Reviewer 3 Report
Comments and Suggestions for Authors
Author Response
Dear Editor,
I have completed my review of the submitted manuscript titled “Circulating Tumour Cells: Origin,
Role, Current Applications and Future Perspectives for Personalized Medicine” and, after careful consideration, I must recommend its rejection.
The iThenticate report revealed a 40% similarity index with other sources, primarily from the paper by Strati et al., Cancers, 2023 (PMID: 37046848). Some paragraphs appear to be directly copied and pasted into the current study, specifically in the Introduction, Result 3, Result 4, Result 5an Result 6 sections. The overlap is mainly found within the main text, raising significant concerns about the originality and contribution of this manuscript.
Given the extent of the duplicated content, I believe the manuscript does not meet the originality standards required for publication in Biomedicines. Therefore, I recommend rejecting this submission.
We answer to the latter refereed very sorry since would not have imagined that our manuscript could be considered as "plagiarism".It certainly wasn't a deliberate act, but rather the not great experience in writing reviews (as co-responding author. Effectively I have written only one (in 2017, Rapanotti et al. Cell Death and Discovery doi: 10.1038/cddiscovery.2017.5) and honestly since I have always included the references in every paragraph and every sentence with complete meaning as well as often, the first author's name in the text , I took it for granted that I was reporting other author’s data. It mortifies me and my collaborators, the judgment of the reviewer 3 because the “literally” reporting the results of the manuscripts, that for us represented the attestation of the value and strength of the data we cited, some of which are considered “guidelines for our daily work” , such as Strati et al., Cancers, 2023 (pmid: 37046848), evidences our greatest recognition of other people's work. Last consideration is that a review work does not report novelties, but tries to summarize and exalt them.
Round 2
Reviewer 1 Report
Comments and Suggestions for Authors
The authors added new texts, Table 1 and re-designed Figures 1 and 2 according to the recommendations.
However, this revised version of manuscript includes the long paragraphs with general statements and with only one or none reference (exact scientific result) and repeating data, especially within the new text.
· There was written several times that CTCs own many genetic changes compared to primary tumor, but epigenetic changes were omitted.
· Rows 108-112 – add the reference(s) to the second model.
· Table 1. – add two columns, one with result of study (for example as CTC count or xxx expression) and the second one with clinical aspects of result (as prognosis of early cancer…).
· Not only single-cell analysis platforms (CTC heterogeneity!), but also multi-omics platforms improve sensitivity and specificity of CTC detection and molecular characterization.
· Avoid repeating of data and re-arrange the text more logically for better readability (maybe sub-paragraphs?).
· Edit typos and “unusual” formulations!
I recommend the second major revision.
Comments on the Quality of English LanguageEdit typos and “unusual” formulations.
Author Response
Answe to Reviewer 1
The authors added new texts, Table 1 and re-designed Figures 1 and 2 according to the recommendations. However, this revised version of manuscript includes the long paragraphs with general statements and with only one or none reference (exact scientific result) and repeating data, especially within the new text.
- We thank the reviewer for his/her observation and added 41 further references. Of them, 38 are original research paper and 3 are reviews.
There was written several times that CTCs own many genetic changes compared to primary tumor, but epigenetic changes were omitted.
- In accordance with the reviewer suggestion we added a paragraph (line 470-504), briefly describing the epigenetic modulation of CTCs (as single CTCs or CTCs cluster) found in different type of tumors. However, we think the topic “epigenetic changes” as already considered for “circulating stem cells” topic should be worthy of a stand-alone review, especially taking into account thatis well-known the epigenetic machinery association to cancer progression and metastasis, but it is unclear whether epigenetic marks in CTCs are clinically relevant biomarkers.
We added 4 more references citing original research paper and one review for a better comprehension of the epigenetic landscape and clinical relevance of CTCs methylation analysis, which is out of our expertise and purpose of the present manuscript. Table 1 has been changed accordingly.
Rows 108-112 – add the reference(s) to the second model.
We added three references: 2 original research papers and 1 previous cited review.
Table 1. – add two columns, one with result of study (for example as CTC count or xxx expression) and the second one with clinical aspects of result (as prognosis of early cancer…).
We thank the reviewer for her/his suggestion and added two columns to the table. We did not detail the gene(s) analyzed to avoid overloading the table output.
Not only single-cell analysis platforms (CTC heterogeneity!), but also multi-omics platforms improve sensitivity and specificity of CTC detection and molecular characterization.
We agree with the reviewer. In accordance with his/her observation we added a brief paragraph (line 464 to 469) and two additional references.
Avoid repeating of data and re-arrange the text more logically for better readability (maybe sub-paragraphs?).
We organized the text in sub-paragraph and deleted the repeated results. To avoid excessive repetition of concepts, we also checked the sequence of references introduced in the text.
Edit typos and “unusual” formulations!
We have corrected the typo mistakes and try to reformulate possible “unusual” formulations.
Reviewer 2 Report
Comments and Suggestions for Authors
- I appreciate the efforts the authors have made to address my concerns.
Author Response
Reviewer 2
I appreciate the efforts the authors have made to address my concerns.
We are grateful that you appreciated our efforts and are committed to "accommodate" your thoughtful change suggestions.
Reviewer 3 Report
Comments and Suggestions for Authors
Overall, the manuscript is now well-written and acceptable. However, there are a few minor issues that require attention.
1. I noticed that the abbreviation 'NSCLC' was introduced in Line 302 without prior definition, while the full term 'non-small cell lung cancer' was only provided later in Line 333. It is crucial to define abbreviations at their first occurrence to ensure clarity for all readers. The authors should review the manuscript for any similar instances and make the necessary corrections to maintain consistency and clarity throughout the text.
2. In Line 368, the sentence ends with a comma before the citation '[35],' which suggests that the sentence may be incomplete or that the comma is unnecessary. A similar issue is observed in Line 450. I recommend reviewing these parts of the manuscript to determine whether the commas should be removed or if the sentences need to be completed. It would also be beneficial to check the entire main text for any similar occurrences.
Author Response
Reviewer 3
Overall, the manuscript is now well-written and acceptable. However, there are a few minor issues that require attention.
- I noticed that the abbreviation 'NSCLC' was introduced in Line 302 without prior definition, while the full term 'non-small cell lung cancer' was only provided later in Line 333. It is crucial to define abbreviations at their first occurrence to ensure clarity for all readers. The authors should review the manuscript for any similar instances and make the necessary corrections to maintain consistency and clarity throughout the text.
- In Line 368, the sentence ends with a comma before the citation '[35],' which suggests that the sentence may be incomplete or that the comma is unnecessary. A similar issue is observed in Line 450. I recommend reviewing these parts of the manuscript to determine whether the commas should be removed or if the sentences need to be completed. It would also be beneficial to check the entire main text for any similar occurrences.
We have added the right definition "non-small cell lung cancer" in extenso, corrected the typo mistakes, the unnecessary commas and try to reformulate possible “unusual” formulations into the entire text.
Round 3
Reviewer 1 Report
Comments and Suggestions for Authors
Corrections at the second revised version of this manuscript make it more complex and readable. It represent current knowledge in CTC clinical utility mostly from the point of view of geneticists.
In addition to the new mutations, epigenetic alterations in CTCs resulting in gene expression changes, which also contribute to the forming of metastatic features of cancer cells including stemness-like characteristics. Therefore, I do not understand your opinion that epigenetic changes are more essential for circulating stem cells, not for CTCs generally. It is possible that clinical importance of epigenetic biomarkers is not clear yet, but similarly, it is not known whether exclusively CTCs with stemness-like features are able to form metastasis.